# Utility of suPAR and NGAL for AKI Risk Stratification and Early Optimization of Renal Risk Medications among Older Patients in the Emergency Department

**DOI:** 10.3390/ph14090843

**Published:** 2021-08-25

**Authors:** Anne Byriel Walls, Anne Kathrine Bengaard, Esben Iversen, Camilla Ngoc Nguyen, Thomas Kallemose, Helle Gybel Juul-Larsen, Baker Nawfal Jawad, Mads Hornum, Ove Andersen, Jesper Eugen-Olsen, Morten Baltzer Houlind

**Affiliations:** 1Department of Drug Design and Pharmacology, University of Copenhagen, 2100 Copenhagen, Denmark; abw@sund.ku.dk (A.B.W.); anne.kathrine.pedersen.bengaard.02@regionh.dk (A.K.B.); camilla_ngoc@hotmail.com (C.N.N.); 2The Capital Region Pharmacy, 2730 Herlev, Denmark; 3Department of Clinical Research, Copenhagen University Hospital—Amager and Hvidovre, 2650 Copenhagen, Denmark; esben.iversen1@hotmail.com (E.I.); thomas.kallemose@regionh.dk (T.K.); helle.gybel.jull-larsen@regioh.dk (H.G.J.-L.); baker.jawad@regionh.dk (B.N.J.); ove.andersen@regionh.dk (O.A.); jespereugenolsen@gmail.com (J.E.-O.); 4Department of Clinical Medicine, University of Copenhagen, 2200 Copenhagen, Denmark; mads.hornum@regionh.dk; 5Emergency Department, Copenhagen University Hospital—Amager and Hvidovre, 2650 Hvidovre, Denmark; 6Department of Nephrology, Copenhagen University Hospital—Rigshospitalet, 2100 Copenhagen, Denmark

**Keywords:** acute kidney injury, early biomarker, plasma neutrophil gelatinase-associated lipocalin, soluble urokinase plasminogen activator receptor, medication optimization, older patients, emergency department

## Abstract

Diagnosis of acute kidney injury (AKI) based on plasma creatinine often lags behind actual changes in renal function. Here, we investigated early detection of AKI using the plasma soluble urokinase plasminogen activator receptor (suPAR) and neutrophil gelatinase-sssociated lipocalin (NGAL) and observed the impact of early detection on prescribing recommendations for renally-eliminated medications. This study is a secondary analysis of data from the DISABLMENT cohort on acutely admitted older (≥65 years) medical patients (*n* = 339). Presence of AKI according to kidney disease: improving global outcomes (KDIGO) criteria was identified from inclusion to 48 h after inclusion. Discriminatory power of suPAR and NGAL was determined by receiver-operating characteristic (ROC). Selected medications that are contraindicated in AKI were identified in Renbase^®^. A total of 33 (9.7%) patients developed AKI. Discriminatory power for suPAR and NGAL was 0.69 and 0.78, respectively, at a cutoff of 4.26 ng/mL and 139.5 ng/mL, respectively. The interaction of suPAR and NGAL yielded a discriminatory power of 0.80, which was significantly higher than for suPAR alone (*p* = 0.0059). Among patients with AKI, 22 (60.6%) used at least one medication that should be avoided in AKI. Overall, suPAR and NGAL levels were independently associated with incident AKI and their combination yielded excellent discriminatory power for risk determination of AKI.

## 1. Introduction

Older people (≥65 years) represent a large and growing demographic worldwide [1,2]. In 2018 alone, approximately 465,000 older people in Denmark were admitted to an emergency department (ED) [3,4]. Acute kidney injury (AKI) occurs in 3–12% of hospitalized patients and is associated with an increased risk of medication-related toxicity, prolonged hospitalization and mortality [5,6,7,8]. The incidence of AKI is particularly high among older patients [9], who are characterized by multiple comorbid conditions that contribute to AKI development [10,11]. Increasing age is also associated with lower baseline glomerular filtration rate (GFR), which predisposes older patients to develop clinically relevant AKI [9,12]. Polypharmacy is common among older patients [13,14] and creates an additional risk in patients at risk for AKI because approximately 40% of all medications are nephrotoxic or require dose adjustment according to estimates of renal function [15]. Epidemiologic studies have identified medication toxicity as a contributing factor in 15–25% of patients with AKI [16,17]. Examples of common nephrotoxic medications that may contribute to AKI include non-steroidal anti-inflammatory drugs (NSAIDs) and renin–angiotensin–aldosterone system (RAAS) inhibitors [18,19]. The combination of age-related changes in kidney function, multiple comorbidities and exposure to polypharmacy with potential nephrotoxic medications is likely responsible for the high rate of AKI among older patients.

AKI involves complex pathophysiology and treatment is largely supportive [20]. AKI may develop prior to hospitalization and go undetected until routine blood samples including creatinine have been performed as a part of standard care [21]. However, increases in plasma creatinine due to AKI often lag 48–72 h behind the onset of injury, resulting in a delayed diagnosis [22,23]. Early detection of AKI at hospital admission may lead to earlier interventions to minimize risk factors or restrict medications that are contributing to AKI [24].

Previous studies have suggested the systemic inflammatory biomarker soluble urokinase plasminogen activator receptor (suPAR) as an early biomarker for detection of AKI [25,26,27,28,29]. suPAR is a signaling glycoprotein thought to be involved in kidney disease pathogenesis [27]. Hayek et al. recently showed that elevated suPAR is associated with increased risk of developing AKI in patients undergoing coronary angiography or cardiac surgery and in patients admitted to the intensive care unit [27]. Some have proposed that suPAR itself may cause kidney disease by damaging renal podocytes [30,31]. However, the applicability of suPAR in predicting AKI among older patients in the ED remains unclear. Another novel biomarker suggested for early detection of AKI is neutrophil gelatinase-associated lipocalin (NGAL) [32,33,34]. NGAL is a member of the lipocalin family of proteins, which is expressed and secreted from renal tubular cells at low concentrations. NGAL is produced in the kidney after ischemic or nephrotoxic injury [35,36,37], and various studies have demonstrated a rise in NGAL 24–36 h before an increase of creatinine is observed [24,38]. Although AKI is common among older patients, there is still a lack of knowledge of the predictive value of using suPAR, NGAL or the combination of suPAR and NGAL for early identification of AKI in older acutely hospitalized patients. The aims of this study are to assess the clinical utility of suPAR and NGAL as early markers of AKI and to quantify the number of renal risk medications that should be dose adjusted or paused in patients presenting with AKI.

## 2. Results

### 2.1. Patient Characteristics and Incidence of AKI 

The original study included 369 patients. Due to the absence of pNGAL value at inclusion, 29 patients were excluded. Further, one patient was excluded due to chronic liver injury, resulting in a total of 339 patients for this study. Patient characteristics for the final study population (*n* = 339) are shown in Table 1. Among included patients, 63% were females, and the median age was 78 years. In median, patients used three renal risk medications. According to KDIGO criteria, 33 (9.7%) patients developing AKI were identified with AKI between inclusion and 48 h after, including 23 with creatinine increased to >1.5 times baseline and 10 patients with creatinine increased by >26.5 µmol/L. Of the 33 patients who developed AKI, 21 patients developed AKI stage 1, while 12 patients developed AKI stage ≥ 2. Compared to patients without AKI, patients who developed AKI had significantly higher Fi-OutRef, creatinine, cystatin C, CRP, IL6 and length of stay, as well as higher change in creatinine and eGFR from admission to discharge (all *p* ≤ 0.01) (Table 1).

### 2.2. Correlations of suPAR, NGAL and eGFR

There was significant correlation between eGFR and levels of suPAR and NGAL (r = −0.35 and −0.53, respectively, both *p* < 0.001) (Appendix A Figure A1a,b). There was also significant correlation between suPAR and NGAL (r = 0.36, *p* < 0.001) (Appendix A Figure A1c). 

### 2.3. SuPAR and NGAL Levels in Patients Developing AKI 

Compared to patients without AKI, those patients who developed AKI had a significantly higher median suPAR (5.8 ng/mL vs. 4.8 ng/mL, *p* < 0.001 (Figure 1a)) and higher median NGAL (229 ng/mL vs. 105 ng/mL, *p* < 0.001 (Figure 1b). Median suPAR was 5.8 (IQR 4.8–9.0) for patients with AKI stage 1 and 5.9 (IQR 4.5–8.7) for patients with AKI stage 2 (*p* = 0.68). Median NGAL was 157 (IQR 123–267) ng/mL for patients with AKI stage 1 and 389 (IQR 280–493) ng/mL for patients with AKI stage 2 (*p* = 0.007).

### 2.4. Risk Prediction for AKI by suPAR and NGAL 

The discriminatory power of suPAR, NGAL or their combination for determining AKI are shown in Table 2 and Figure 2. As individual biomarkers for the detection of AKI, suPAR yielded an AUC of 0.69 with an optimal cut-off of 4.26 ng/mL, and NGAL yielded an AUC of 0.78 with an optimal cut-off of 139.5 ng/mL. No significant difference was found between AUC for suPAR and AUC for NGAL (*p* = 0.117). The interaction of suPAR and NGAL yielded an AUC of 0.80, which was significantly higher than AUC for suPAR alone (*p =* 0.0059) but not for NGAL alone (*p =* 0.689) (Figure 2). The addition of CRP or CRP + IL6 did not significantly improve AUC for any models (*p* ≥ 0.108) (Appendix A Figure A2). However, the addition of CRP to suPAR improved the AUC to 0.76, which is considered to be acceptable discriminatory power.

Cut-off values for combinations of suPAR and NGAL from the 2-variable interaction model show a dependency between the variables with lower values of NGAL requiring larger suPAR values (9.6 ng/mL suPAR at NGAL 2.6 ng/mL) and larger values of NGAL requiring smaller suPAR values (0.5 ng/mL suPAR at 205 ng/mL NGAL) (Figure 3). Further, the 3-variable interaction model show the dependency between suPAR and NGAL values at the cut-off was notably larger with eGFR < 60 mL/min/1.73 m^2^ (Appendix A Figure A3).

### 2.5. Renal Risk Medications in Patients Developing AKI

Among those with AKI, 20 (60.6%) patients used at least one medication that should be avoided in AKI, and 7 (21.2%) patients used two or more of these medications (Table 3).

## 3. Discussion 

### 3.1. Main Findings

In this study, we assess the applicability of suPAR and NGAL as early biomarkers of AKI in older acutely hospitalized patients. In total, 9.7% of the study group developed AKI within 48 h after study inclusion. Concentrations of suPAR and NGAL were correlated with AKI severity and reduced eGFR. ROC analysis for suPAR and NGAL yielded AUCs of 0.69 and 0.78 and cutoff values at 4.26 ng/mL and 139.5 ng/mL, respectively. The combination of suPAR and NGAL yielded an AUC of 0.80, which was significantly higher than for suPAR alone (*p* = 0.032). Among patients with AKI, 22 (60.6%) used at least one medication that should be avoided in patients with AKI. 

### 3.2. AKI in Older Acutely Hospitalized Patients 

Older patients are more susceptible to developing AKI due to multimorbidity [10,11], physiological reduction in GFR [9,12] and polypharmacy [13,14]. The prevalence of AKI in our study is 9.7%, which is slightly higher than what has been reported in similar studies [5,7,12]. This difference likely reflects the demographic composition of older medical patients predisposed to developing AKI [12]. Patients with AKI were hospitalized longer than those without AKI, which is in accordance with previous studies [7,8]. We also observed that the inflammatory biomarkers CRP, IL6 and *TNF-α* were higher among patients who developed AKI compared to those who did not, which highlights the role of severe infection in the pathogenesis of AKI [8,39]. Patients with AKI exhibited significantly higher median plasma levels of suPAR and NGAL compared to patients without AKI (Figure 1). Plasma suPAR and NGAL levels were also inversely correlated with baseline eGFR (Figure A1), which supports previous literature demonstrating the connection between these biomarkers and kidney function [25,27,29]. The associations with suPAR may indicate the role of suPAR in systemic inflammation, which is expected to be elevated in our study group. They may also indicate a value for suPAR in predicting AKI, which has previously been demonstrated in a variety of patient populations including those undergoing cardiac surgery, admitted to an intensive care unit or infected with COVID-19 [25,26,27,28,29,40]. However, suPAR appears to be unrelated to AKI severity, while plasma NGAL increased significantly with AKI severity, similar to findings by Soto et al. [32]. In future studies, more sophisticated prediction models may be developed using NGAL cutoff values for different degrees of AKI severity.

### 3.3. Plasma suPAR and NGAL 

Several studies have suggested plasma suPAR as a biomarker for early detection of AKI. Our findings demonstrate that suPAR has a sensitivity of 94%, specificity of 40% and discriminative ability (AUC) of 0.69 for the development of AKI at a cutoff of 4.26 ng/mL. These findings are compatible with a similar study in patients undergoing cardiac surgery, which reported an AUC of 0.65 for the development of AKI at a suPAR cutoff value of 2.45 ng/mL [29]. Rasmussen et al. also investigated the discriminatory power of suPAR for AKI in patients undergoing cardiac surgery and reported an AUC of 0.60 [40]. The difference in cutoff values between these studies and our own may indicate a higher overall inflammatory state among patients in our study. In contrast, a study conducted in hospitalized patients with COVID-19 found an AUC of 0.75 at a cutoff value of 4.60 ng/mL [28], likely reflecting the high inflammatory burden of COVID-19.

Several previous studies also support the use of plasma NGAL for early AKI detection [34]. We found that NGAL has a sensitivity of 76%, specificity of 67% and discriminative ability of 0.78 for the development of AKI at a cutoff of 139.5 ng/mL. A multicenter study in the USA by Shapiro et al. assessed the predictive value of pNGAL in 1015 patients (average age 59) in the ED with suspected sepsis and found that pNGAL was 96% sensitive and 51% specific with an AUC of 0.78 for the development of AKI at a cutoff of 150 ng/mL [23]. Using the same pNGAL cutoff value, a study in Portugal by Soto et al. among 616 patients (average age 59) admitted to the ED reported an AUC between 0.77 and 0.82 for the development of AKI depending on when NGAL was measured [32]. Finally, a multicenter study in Italy by Di Somma et al. among 665 patients (average age 74) admitted to the ED reported an optimal pNGAL cutoff of 137 ng/mL, resulting in an AUC between 0.79 and 0.84, depending on AKI definition [41]. Overall, our reported AUC of 0.78 at a cutoff of 139.5 ng/mL is highly comparable to these other studies in similar patient populations. A recent meta-analysis reviewing NGAL as predictor for AKI reported an overall AUC of 0.74 at a cutoff of 165 ng/mL for all available studies [34], which is largely compatible with our findings. Results from the same meta-analysis highlighted that urinary NGAL measured in urine is also a robust biomarker for detecting AKI [34]. Measurement of urinary NGAL is non-invasive and should be considered in settings where measurement of plasma NGAL requires additional blood draws.

Since November 2013, suPAR but not NGAL has been routinely measured in all patients admitted to the ED at our hospital. We have previously shown that suPAR can be used for overall risk stratification and safe early discharge [25]. During weekdays, suPAR is measured once or twice per day, and results are available on average 16 h (range 2–74 h) after admission. Therefore, suPAR values are often not reported before clinical decisions are made for acute admissions. Quicker turnaround times are required if suPAR or NGAL should be used for early AKI risk stratification in the ED. One solution is to analyze both biomarkers using point-of-care or turbidimetric assays. It may also be useful for patients with elevated suPAR or NGAL during a previous admission to be flagged in the electronic patient record for future clinical encounters. A recent study by Mossanen et al. suggested that the combination of suPAR and NGAL may strengthen the prediction of AKI [29]. We found that plasma NGAL alone yielded an AUC of 0.78 for the development of AKI, while the addition of suPAR improved the AUC to 0.82. Such a change in discriminatory ability may not be clinically relevant, but results from Iversen et al. suggest that elevated suPAR at hospital admission reflects increased long-term risk of AKI after hospital discharge [25], maybe because suPAR in itself is involved in the pathogenesis of AKI [27]. Therefore, perhaps NGAL is more useful for predicting impending AKI in an acute setting whereas suPAR is more useful for predicting future AKI after discharge. In clinical settings where suPAR is already implemented as a standard biomarker, we suggest that suPAR in combination with CRP should be utilized for AKI risk stratification.

### 3.4. Optimization of Medication Prescribing 

In total, 33 patients in our study developed AKI within 48 h of ED admission. These patients used a median of eight medications, approximately 40% of which are considered renal risk medications [15]. Among patients who developed AKI, 20 (60.6%) used ≥1 renal risk medication that should be avoided in patients with AKI, with opioids being the most common example. Given the known interactions between AKI and renal risk medications, early detection of AKI is essential for limiting the effects of nephrotoxic medications as well as reducing the dose of medications excreted by the kidneys. Results from this study indicate that plasma suPAR and NGAL can be used to screen patients for risk of developing AKI. A positive screen for high risk of AKI can prompt healthcare practitioners to perform a comprehensive medication review to identify renal risk medications that should be discontinued, dose-adjusted or monitored during hospitalization. We believe the use of routine biomarkers in combination with automated screening precautions would result in faster interventions to optimize medication prescribing among acutely hospitalized older patients at high risk for developing AKI.

### 3.5. Strengths and Limitations

The primary strength of this study is its applicability to a daily clinical challenge in the ED. Acutely hospitalized patients, and particularly those who are older with multimorbidity, are at elevated risk for developing AKI, yet there are currently no reliable tools for quickly identifying which patients are at the highest risk. Our study identifies screening tools that are both efficient and easily implemented given the time constraints of the ED. This study also has some limitations. First, we did not have access to creatinine values prior to admission. Second, our definition of AKI is limited to plasma creatinine and does not account for urine output. Third, both suPAR and NGAL can be affected by other clinical factors which may confound their association with thendevelopment of AKI. We attempted to account for these factors by excluding patients with chronic liver disease, but we could not account for subclinical conditions such as low-grade inflammation or asymptomatic infection. Fourth, we used Renbase^®^ to determine prescribing recommendations for renal risk medications, but there may be discrepancies between Renbase^®^ and other medication databases. Finally, the study is a single center study, and results should be confirmed in larger multicenter studies.

## 4. Materials and Methods

### 4.1. Setting

This study is a secondary analysis of data from the Disability in Older Medical Patients (DISABLMENT) cohort, which aimed to investigate the ability of physical performance measures and biomarkers to predict adverse health events in older patients after acute medical hospitalization and one year after discharge [42,43]. The study was performed in the Emergency Department (ED) at Hvidovre Hospital, University of Copenhagen, Denmark between July 2012 and September 2013.

### 4.2. Design and Participants

The original DISABLMENT [42,43] study included 369 older medical patients acutely admitted to the ED. The inclusion criteria were age ≥65 years and acutely admitted for a medical illness to the ED. The exclusion criteria were inability to cooperate, an inability to communicate in Danish, a cancer diagnosis or terminal disease, patient isolation, admission to an intensive care unit or imminent discharge hindering interview and physical testing. Using a computer-generated list, eligible patients were included using random sampling based on their social security number, as it was not possible to include all eligible patients due to assessment resources [42,43]. For the current study, patients were also excluded if the NGAL value was not measured or if they had a chronic liver injury (if prescribed in electronic patient record).

### 4.3. Ethical Statement

The original DISABLMENT cohort was conducted in accordance with the Declaration of Helsinki. Signed informed consent was obtained from all participants, and the study was approved by the Danish Data Protection Agency (0159 HVH-2012-005) and the Research Ethics Committees for the Capital Region (H-1-2011-167).

### 4.4. Patient Demographic, Length of Stay and Mortality 

Patients’ age and gender were recorded at admission. Patients were included in the study within 24 h after admission. Patient demographic information as physical parameters including weight and height were measured during this time. Data of cardiovascular disease and diabetes were identified by ICD-10 diagnosis codes or ATC medication codes in each patient’s medical record within 10 years before inclusion in the study as described in Juul-Larsen et al. 2019 [44] Data regarding length of stay and 30-days mortality were obtained from the patient’s electronic health records. Patients’ frailty index (FI-OutRef) representing cumulative organ dysfunction, calculated as number of laboratory results outside of reference interval for 17 standard biomarkers, collected at admission: C-reactive protein (CRP), leucocytes, neutrophils, haemoglobin, mean corpuscular haemoglobin concentration (MCHC), mean corpuscular volume (MCV), thrombocytes, creatinine, blood urea nitrogen (BUN), sodium, potassium, albumin, alanine aminotransferase (ALAT), alkaline phosphatase, lactate dehydrogenase, (LDH), bilirubin and factors II, VII and X [45,46].

### 4.5. Timepoints for Measuring Biomarkers and Calculation of Baseline Plasma Creatinine 

Patients’ plasma creatinine, NGAL and suPAR value at inclusion (day 0) was obtained from the samples stored in a biobank. Creatinine values were measured repeatedly during hospitalization. Creatinine values at 24 h (day 1) and 48 h (day 2) after inclusion were obtained from the electronic patient record. The lowest measured creatinine value from admission to discharge, obtained from the electronic patient record or biobank, was defined as baseline (Appendix A
Figure A4). Discharge creatinine was defined as the last measurement during admission.

### 4.6. Determination of Biomarkers

Blood samples were obtained at inclusion and stored at −80 °C in a Biobank at Copenhagen University Hospital in Hvidovre. Creatinine was measured by absorption photometry on a Roche Cobas^®^ c 8000 701/702 with a module instrument using the Roche Creatinine Plus version 2 IDMS-traceable assay (coefficient of variation 1.5%). NGAL was measured on a Roche Cobas^®^ c 8000 501/502 with the NGAL Test™ using particle-enhanced turbidimetric immunoassay (PETIA) (Bioporto^®^, Hellerup, Denmark) (coefficient of variation 3.7%). suPAR was measured using an enzyme-linked immunosorbent assay (suPARnostic^®^ Auto Flex ELISA) (ViroGates A/S, Birkerød, Denmark) (coefficient of variation 3%) [43]. C-reactive protein (CRP) was measured by turbidimetric immunoassay on a Roche Cobas^®^ 6000 platform in (Roche Diagnostic, Mannheim, Germany) [45]. Cystatin C was also measured on a Roche Cobas^®^ c 8000 701/702 with a module instrument using the Roche Cystatin C Tina-quant generation 2 particle-enhanced immunonephelometric assay [45]. IL-6 and TNFα concentrations were measured on a Luminex^®^ 200 platform (Luminex, Austin, TX, USA) using the Milliplex Human Cytokine/Chemokine Magnetic Bead Panel (Millipore, Billerica, MA, USA) as described in Klausen et al. 2017 [43]. 

### 4.7. Estimated Glomerular Filtration Rate

The chronic kidney disease epidemiology collaboration (CKD-EPI) equation based on creatinine (CKD-EPI_Cr_) was used to estimate eGFR without adjustment for race [47]. Estimated GFR was calculated using the creatinine level at which suPAR and NGAL was measured at inclusion. 

### 4.8. Medication

Patients’ medication data were obtained from the Shared Medication Card Online, which records all prescriptions obtained by patients at a primary pharmacy [45]. This study only included medications for systemic use. Medication retrieved from a pharmacy within 4 months of hospital admission were included [45]. Prescriptions with end dates prior to admission or start dates after admission were excluded. Prescribed daily dose was calculated from dosing strength and frequency. The maximum daily dose was used if the medication was prescribed “as needed” [45]. 

According to Renbase^®^, renal risk drugs are defined as drugs that should either be avoided or dose-adjusted according to GFR [48]. Apart from the median value of renal risk drugs being used, this study is limited to a list of selected renal risk medications; metformin (A10BA02), NSAIDs (M01A (except of M01AX)), opioids which are further limited to tramadol (N02AX02), codeine (R05DA04) and morphine (N02AA01), whereas angiotensin-converting enzyme inhibitors (ACEIs) (C09AA) and angiotensin II receptor blockers (ARBs) (C09CA) were included for all drugs within the groups. These drugs should be avoided in the presence of AKI [49].

### 4.9. Outcomes

In this study, we have three outcomes to address the applicability of suPAR and NGAL as a prognostic kidney biomarker for AKI: (1) the accuracy of suPAR in predicting AKI between inclusion and 48 h after, (2) the accuracy of NGAL in predicting AKI between inclusion and 48 h after and (3) the accuracy of suPAR in combination with NGAL in predicting AKI between inclusion and 48 h after. 

AKI is defined by the Kidney Disease: Improving Global Outcomes (KDIGO) Work Group criteria as an increase in creatinine to ≥1.5 times baseline or increase in creatinine by ≥0.3 mg/dL (≥26.5 μmol/L) within 48 h. The lowest measured creatinine value during hospitalization was defined as baseline creatinine. We identified patients with AKI from inclusion and within 48 h. Severity of AKI is classified according to the KDIGO criteria. Stage 1 is defined by an increase of 1.5–1.9 times baseline or an increase in creatinine by ≥26.5 μmol/L. An increase of 2.0–2.9 times baseline is defined as stage 2, and stage 3 is defined by an increase of 3.0 times baseline or more, or an increase in creatinine by (≥353.6 μmol/L) [50].

### 4.10. Statistical Analysis

Data were processed using Microsoft Excel XLSTAT. Continuous variables are given as median with interquartile range (IQR), and discrete variables are given as number with percent of patients. Continuous variables were compared by Mann–Whitney U test; tests for biomarkers, creatinine change and eGFR change were adjusted for multiple testing by Bonferroni correction by upscaling p-values with number of tests. Correlation between continues variables were estimated by Pearson correlation coefficient, and tested against a correlation of 0. The discriminatory value of NGAL and suPAR in relation to AKI was analyzed by receiver operating characteristic (ROC) analysis. Single-term models for suPAR (model 1) and NGAL (model 2), an interaction model with NGAL and suPAR included (2-variable interaction) (model 3) and an interaction model with NGAL, suPAR and eGFR (>60/<60 mL/min/1.73 m^2^ at inclusion) included (3-variable interaction) were analyzed. Additionally, versions of model 1–3 with the addition of CRP and IL6 were also analyzed. Cut-off values from the ROC analysis were based on maximizing the Youden index. Models were fitted as logistic regression models and the linear predictor used as the continues predictor in the ROC analysis, cut-off values were calculated for the linear predictor and afterwards transformed back to specific suPAR and NGAL values. For interaction models, multiple cut-off values for suPAR are given dependent on the NGAL value and vice versa; because of this, the cut-off values are presented graphically. Area under the curve (AUC) is presented with 95% confidence interval (CI) and compared between the models. All analyses were performed using R 3.6.0 [30] with ROC analysis using the pROC r-package [51]. An AUC value of 0.7–0.8 is considered acceptable; 0.8–0.9 is considered excellent and a value more than 0.9 is considered outstanding [52]. A *p*-value of less than 0.05 was considered statistically significant.

## 5. Conclusions

AKI and use of renal risk medications are common among older patients in the ED. We found that suPAR and NGAL levels were independently associated with incident AKI, and the combination of suPAR and NGAL yielded excellent discriminatory power for risk of developing AKI. However, discriminatory power of suPAR and NGAL in combination was not statistically different from NGAL alone. The discriminatory power of suPAR and NGAL in older medical patients was similar to findings in the existing literature with other groups of patients.

## Figures and Tables

**Figure 1 pharmaceuticals-14-00843-f001:**
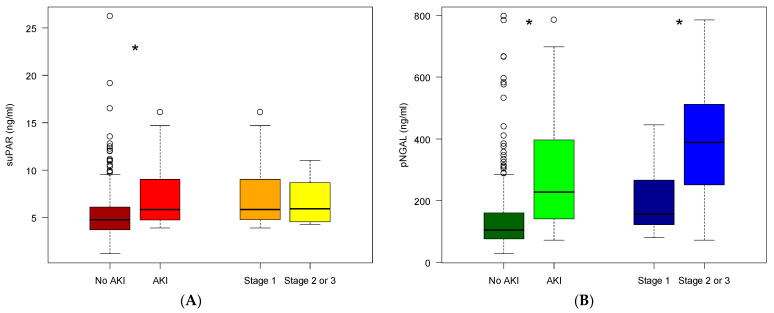
Plasma concentration of suPAR and NGAL at inclusion. (**A**) suPAR values in patients: without AKI (brown), developed AKI (red) within 48 h after inclusion, developed AKI stage 1 (orange), developed AKI stage ≥2 (yellow). (**B**) NGAL values in patients: without AKI (dark green), developed AKI (light green) within 48 h after inclusion, developed AKI stage 1 (dark blue), developed AKI stage ≥2 (light blue). The horizontal lines show minimum and maximum values of calculated non-outlier values; open circles indicate outlier values (* *p* < 0.05).

**Figure 2 pharmaceuticals-14-00843-f002:**
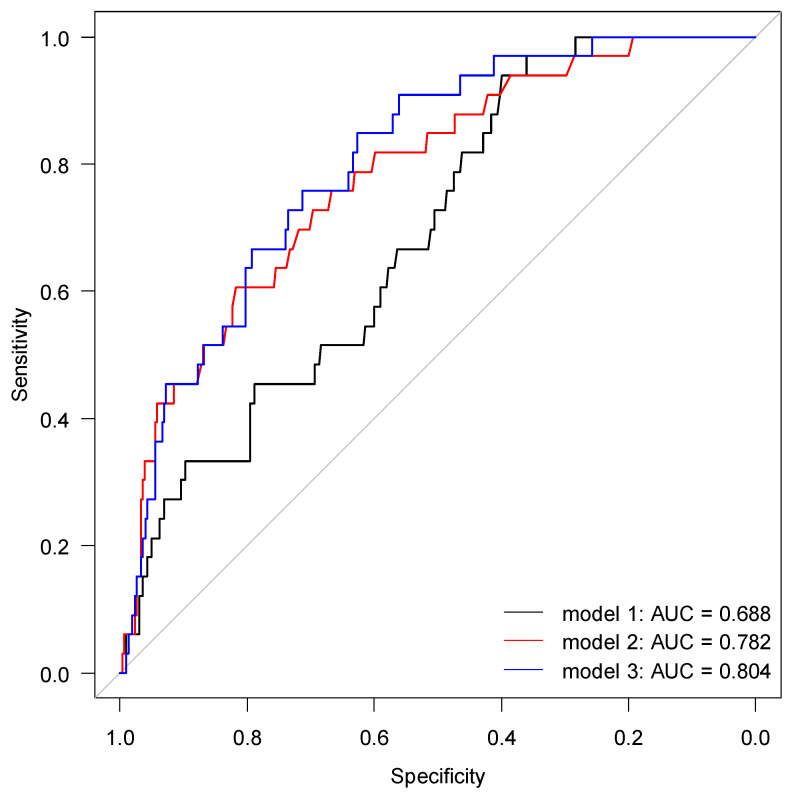
Receiver operating characteristic curve for predicting AKI. Model 1 includes suPAR; model 2 includes NGAL; and model 3 includes interaction between suPAR and NGAL (2-variable interaction).

**Figure 3 pharmaceuticals-14-00843-f003:**
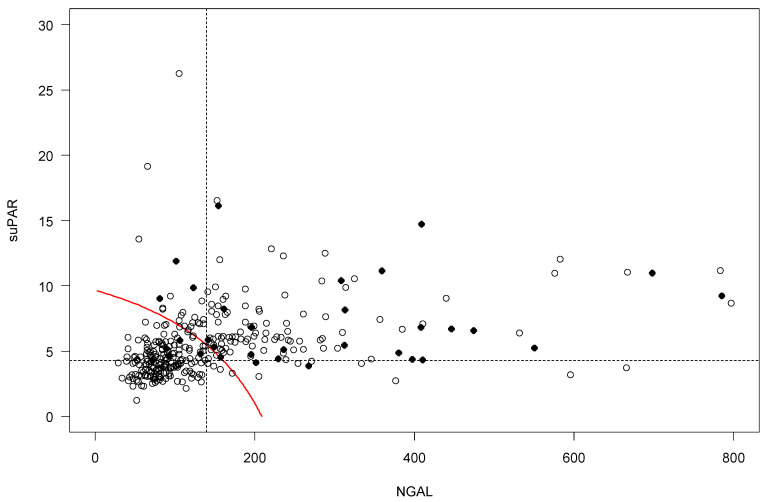
Two-biomarker cut-off approach with suPAR and NGAL (two-variable interaction). The dotted lines represent the cut-off values for NGAL and suPAR set to 139.5 ng/L and 4.26 ng/L, respectively. The red curve illustrates cut-off values for the combinations of suPAR and NGAL.

**Table 1 pharmaceuticals-14-00843-t001:** Patient characteristics for all included patients, patients with and without AKI.

Variable	All Patients		Patients with AKI		Patients without AKI	
	N	Value	N	Value	N	Value
**Demographics**						
Age years, median (IQR)	339	77.6 (70.6; 84.4)	33	75.9 (72.3; 83.0)	306	77.9 (70.5; 84.5)
Female, *n* (%)	-	212 (62.5)	-	25 (75.8)	-	187 (61.1)
Body-mass index, median (IQR)	304	25.1 (22.3; 28.8)	26	24.8 (20.7; 28.9)	278	25.1 (22.5; 28.8)
Hospitalization-days, median (IQR)	339	2 (1; 6)	33	7 (4; 13)	306	2 (1; 5)
30-day morality, *n* (%)	339	12 (3.5)	33	3 (9.1)	306	9 (2.9)
**Comorbidities**						
Cardiovascular disease (%)	-	113 (33.3)	-	12 (36.4)	-	101 (33.0)
Diabetes (%)	-	57 (16.8)	-	5 (15.2)	-	52 (17.0)
**Medication**						
Total number of medications, median (IQR)	339	6 (3; 9)	33	8 (4; 12)	306	6 (3; 9)
**Biomarkers ***						
Creatinine µmol/L, median (IQR)	339	84.3 (66.2; 105.4)	33	120.8 (91.1; 169.5)	306	83.0 (65.4; 100.2)
Cystatin C mg/L, median (IQR)	339	1.21 (0.95; 1.60)		1.69 (1.26–2.56)	306	1.17 (0.94; 1.56)
eGFR mL/min/1.73 m^2^, median (IQR)	339	65.6 (48.2; 81.9)	33	39.1 (26.7; 59.2)	306	67.4 (50.7; 82.3)
CRP-µg/mL, median (IQR)	314	15.5 (3.0; 63.7)	33	67.0 (22.3; 120.3)	281	14.0 (3.0; 53.4)
IL6-pg/mL, median (IQR)	336	4.6 (1.9; 13.3)	33	9.8 (3.6; 30.4)	303	4.3 (1.8; 11.1)
*TNF-α*–pg/mL, median (IQR)	336	7.4 (5.1; 107)	33	10.1 (6.7; 14.9)	303	7.3 (4.9; 10.5)
FI-OutRef, median (IQR)	314	5 (3; 7)	33	7 (6; 8)	282	5 (3; 7)
**Change in creatinine and eGFR ****						
Δ_creatinine_ inclusion to discharge	339	−1.0 (−9.0:7.0)	33	−33.0 (−57.0:−13.0)	306	0.0 (−7.0:7.0)
Δ_eGFR_ inclusion to discharge	339	1.0 (−4.1:7.1)	33	20.4 (4.4:32.2)	306	0.0 (−4.7:4.9)

AKI, acute kidney injury; eGFR, estimated glomerular filtration rate calculated with chronic kidney disease epidemiology collaboration (CKD-EPI) equation based on creatinine; CRP, C-reactive protein; IL-6, interleukin 6; TNFα, tumor necrosis factor alpha. * *p*-values multiplied by seven. ** *p*-values multiplied by two.

**Table 2 pharmaceuticals-14-00843-t002:** Diagnostic accuracy of suPAR, NGAL and the combination of both biomarkers, using optimal cut-off values, for predicting AKI.

	Cutoff	Sensitivity	Specificity	PPV	NPV	AUC (CI 95%)
suPAR (ng/mL)	4.26	0.94	0.40	0.15	0.98	0.69 (0.60–0.77)
NGAL (ng/mL)	139.5	0.76	0.67	0.20	0.96	0.78 (0.70–0.87)
Two-variable interaction	-	0.82	0.73	0.25	0.97	0.82 (0.73–0.90)

Two-variable interaction, includes interaction between suPAR and NGAL.

**Table 3 pharmaceuticals-14-00843-t003:** The table shows the frequency of patients with AKI using selected renal risk drugs that should be avoided.

	AKI (*n* = 33) (%)
Opioids	13 (39.4)
NSAIDs	4 (12.1)
Metformin	4 (12.1)
ACEIs/ARBs	10 (30.3)

AKI, acute kidney injury; NSAIDs, nonsteroidal anti-inflammatory drugs. ACEIs, angiotensin-converting enzyme inhibitors. ARBs, angiotensin II receptor blockers.

## Data Availability

Data available on request due to restrictions. The data presented in this study are not publicly available due to Danish legislation. Request to access the dataset will require an individual inquiry to the Danish Data Protection agency for approval.

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
