# Peer review of "Utility of suPAR and NGAL for AKI Risk Stratification and Early Optimization of Renal Risk Medications among Older Patients in the Emergency Department"

_pharmaceuticals, 2021, doi:10.3390/ph14090843_

Round 1
Reviewer 1 Report
Walls et al. answered the question of the clinical utility of suPAR and NGAL as early markers of AKI and they also assessed the use of renal risk medications that should be avoided or dose adjusted in AKI status. There are some evidence regarding the clinical utility of suPAR and NGAL, however, it is unknown in elderly who present emergent room. The manuscript is well written with scientific manner and the information of the present study is interesting and important for further research.
Author Response
Thank you for your positive response to our manuscript (pharmaceuticals-1307792). Best, Morten Houlind.
Reviewer 2 Report
Dear Dr. Houlind,
your paper is written in a very comprehensive style and offers some very valuable results.
Therefore only some minor aspects/clarifications are to be added:
1) regarding the given renal biomarker (creatinine, eGFR) it seems that pts. with AKI are suffering from pre-existing, impaired renal function (eGFR 39 = CKD GbAx). Thus it would enhance clinical relevance if renal biomarkers at admission to ED, 24 h and 48 hours after admission and at discharge are given; what was the Δ creatinine and ΔeGFR between AKI and non-AKI pts. ad 24 h and 48 h after admission;
2) the performed analysis of suPAR and NGAL was done with stored biobank samples, so somehow in a “retrospective manner”, which nevertheless showed valuable results; the authors also stated that suPAR is routinely measured upon pts. admission to ED: what is the mean turnaround time for suPAR measurement at the hospital and is it available 24/7; is NGAL also routinely available? please comment on these aspects to assess if both markers can be used for “daily-patient-care”;
3) if suPAR is measured routinely in ED: are pts. with elevated suPAR levels “tagged” in the clinic information system so that subsequent departments are informed about the pts. elevated risk for AKI and so that medication can be adopted to pts ` risk profile: please comment
4) as urinary NGAL is also very robust for detection of AKI, but can be acquired non-invasively it would be an idea for future studies
5) are the given inflammatory biomarkers (e.g. CRP,…) statistically different in AKI vs non-AKI pts.; because if so additional adjustment regarding the discriminatory impact of suPAR/NGAL regarding AKI detection should be performed
Author Response
Thank you for your response to our manuscript (pharmaceuticals-1307792).
Please see the attachment.
